

# Metabolite profiling from the fermentation of marine-derived extracts by *Lactobacillus acidophilus* LB

Ha Phuong Hoang, Thi Minh Nguyen, Tuyet Thi Anh Le, Huong Giang Bui, Ngoc Anh Ho, Thu Ngo Thi Hoai and Nhat Huy Chu

Graduate University of Science and Technology, Vietnam Academy of Science and Technology, Ha Noi, Vietnam

Institute of Biotechnology, Vietnam Academy of Science and Technology, Ha Noi, Vietnam

## ABSTRACT

**Background**. *Lactobacillus acidophilus* LB is a probiotic strain with the ability to produce valuable bioactive metabolites through fermentation. Sustainable biomass sources such as *Spirulina platensis*, *Ulva reticulata*, and *Caulerpa lentillifera*, which can also be obtained from agricultural or aquacultural by-products, offer a promising alternative for microbial cultivation, but their effects on the metabolic profile of *L. acidophilus* LB are still unclear.

**Methods**. The study utilized *L. acidophilus* LB (GenBank accession OK398226) cultivated in media containing *Spirulina platensis*, *Ulva reticulata*, and *Caulerpa lentillifera* extracts. High-resolution mass spectrometry (liquid chromatography-high resolution mass spectrometry-quadrupole-time-of-flight, LC-HRMS QTOF) was employed for compound profiling. Principal component analysis (PCA) and Pearson correlation were used to analyze metabolic variations across different culture conditions. The Kruskal-Wallis test assessed statistical differences in metabolite concentrations.

**Results**. Methyl lucidenate Q and 6-Gingerol were the most abundant bioactive compounds detected across samples. PCA revealed that *L. acidophilus* LB in media supplemented with different preparations of *Spirulina platensis*, *Ulva reticulata*, and *Caulerpa lentillifera* was associated with distinct differences in metabolite profiles, leading to clustering patterns. K-means clustering identified three metabolomic groups, with the pellet obtained from *L. acidophilus* LB cultured in medium supplemented with *U. reticulata* seaweed showing a unique chemical profile. Pearson correlation analysis suggested possible biochemical interactions among metabolites, with Auraptenol and Daturametelin B exhibiting a strong positive correlation ($r = 0.99$). The absence of Methyl lucidenate Q in this sample indicates potential enzymatic degradation or metabolic inhibition.

**Conclusion**. This study highlights the impact of *L. acidophilus* LB on metabolite diversity in substrate-driven fermentation systems. The findings suggest microbial interactions modulate metabolite patterns in the fermented supernatants, potentially enhancing pharmacological properties. Future research should focus on optimizing culture conditions to maximize yield and functional validation of identified compounds for therapeutic applications. These insights contribute to the broader field of natural product discovery and marine biotechnology.

Corresponding author
Nhat Huy Chu,
chunhathuy@gmail.com

## INTRODUCTION

*Lactobacillus acidophilus* is a probiotic strain with considerable biotechnological potential, capable of producing functional metabolites such as exopolysaccharides, biosurfactants, and organic acids that are valuable in various industrial applications (*Brzozowski, Bednarski & Dziuba, 2009*; *Maryati, Nuraida & Hariyadi, 2021*). Marine algae and seaweeds are increasingly recognized as key marine biomass resources, offering sustainable and versatile carbon substrates for microbial cultivation and the production of value-added products (*Nguyen et al., 2022*).

*Spirulina platensis* and *Caulerpa lentillifera* extracts have been shown to enhance the growth and metabolic activity of *L. acidophilus*, leading to increased fermentation efficiency and the production of bioactive compounds (*Barros De Medeiros et al., 2022*; *Chadseesuwan, Puthong & Deetae, 2021*; *Hoang et al., 2024*). In contrast, the fermentation potential of *Ulva reticulata* with *L. acidophilus* remains underexplored, despite the species' documented antibacterial, antioxidant, and antidiabetic properties (*Unnikrishnan et al., 2022*; *Djoh et al., 2024*).

Beyond their biological advantages, seaweeds offer important environmental benefits: rapid growth, no requirement for arable land or freshwater, and minimal competition with the human food chain. Moreover, byproducts such as aged or substandard biomass from cultivation and processing can be repurposed as fermentation feedstocks, thereby reducing waste and increasing the value of aquatic production chains (*Znad, Awual & Martini, 2022*; *Pasanda & Kusuma, 2016*).

*Ulva reticulata* contains reducing sugars, proteins, flavonoids, and polyunsaturated fatty acids, notably palmitic acid (*Gomathi & Anna, 2018*). Its extracts show antimicrobial, antioxidant, and antidiabetic activities, varying with extraction solvents such as methanol and chloroform (*Djoh et al., 2024*; *Unnikrishnan et al., 2022*). *Caulerpa lentillifera* is rich in proteins, phenolics, and polysaccharides, exhibiting antioxidant, antimicrobial, and antidiabetic effects (*Nurkolis et al., 2023*), with bioactivity influenced by cultivation and extraction conditions (*Syakilla et al., 2022*). *Spirulina platensis* provides high protein content and key bioactives like phycocyanin and γ-linolenic acid, supporting antioxidant and immunomodulatory functions (*Spínola, Mendes & Prates, 2024*; *Bellahcen et al., 2020*), with composition varying based on cultivation conditions. While the chemical composition and bioactivities of *Spirulina platensis*, *Ulva reticulata*, and *Caulerpa lentillifera* have been well studied, the metabolite profiles resulting from their fermentation with *Lactobacillus acidophilus* LB remain largely unexplored.

Based on the hypothesis that biochemical differences among seaweed species may influence bacterial metabolic responses, we cultured *L. acidophilus* LB in media containing extracts from three distinct seaweed species as the sole carbon source. The objective was to evaluate the ability of these extracts to support bacterial growth and to examine their impact on metabolite profiles in the culture media, with the aim of identifying distinct metabolic patterns, characteristic biomarkers, and sample clustering through statistical analyses.

Principal Component Analysis (PCA) and Pearson correlation are widely used tools for analyzing metabolic differences across samples (*Pearson, 1895*). For this research, these methods were employed to investigate metabolic variation and to identify patterns in the data (*Chatfield & Collins, 1980*; *Cooley & Lohnes, 1971*; *Morrison, 1967*).

In this study, we applied an untargeted metabolomics approach to comprehensively investigate metabolic changes in the culture media after fermentation of *Spirulina platensis*, *Ulva reticulata*, and *Caulerpa lentillifera* with *L. acidophilus* LB. Liquid chromatography-quadrupole-time-of-flight (LC-QTOF) analysis combined with multivariate statistical methods was employed to explore data structure and identify potential bioactive or industrially relevant metabolites.

## MATERIALS AND METHODS

### Materials

The *Lactobacillus acidophilus* LB strain (GenBank accession number OK398226) was preserved in the strain collection of the Environmental Bioremediation Laboratory, Institute of Biotechnology, Vietnam Academy of Science and Technology (VAST). The dry biomass of *Spirulina platensis*, *Ulva reticulata*, and the extract of *Caulerpa lentillifera* were provided by the Algal Biotechnology, Institute of Biotechnology (VAST). For *Caulerpa lentillifera*, the seaweed was directly blended with water at a ratio of 1:1 (w/v), and the mixture was filtered to remove solids, yielding an extract used for the experiments.

### Preparation of media

Media containing *S. platensis* or *U. reticulata* biomass were prepared by adding 3.5% (w/v) algal/seaweed biomass and 2% glucose (w/v) into a phosphate buffer solution (PBS), followed by sterilization to eliminate bacterial contaminants. For *C. lentillifera*, the extract (40% v/v) was added to PBS buffer to create the bacterial growth medium.

In all experiments, de Man, Rogosa and Sharpe (MRS) medium, containing peptone (10 g/L), yeast extract (five g/L), meat extract (10 g/L), glucose (20 g/L), ammonium citrate (two g/L), sodium acetate (five g/L), magnesium sulfate (0.1 g/L), manganese sulfate (0.05 g/L), dipotassium hydrogen phosphate (two g/L), and Tween 80 (one mL/L), served as the medium control. This enabled the comparison of *L. acidophilus* LB growth in traditional media and modified media containing algal or seaweed biomass.

### Evaluation of *L. acidophilus* LB growth in algal/seaweed-containing media (Table S1)

A total of 14 samples were prepared, including six extract samples from *Spirulina platensis*, *Ulva reticulata*, or *Caulerpa lentillifera* fermented with *Lactobacillus acidophilus* LB, six

unfermented extract samples from the same algae, and two control samples of *L. acidophilus* LB cultured in MRS medium. Sample codes were assigned as follows: S. for supernatant (extract), B. for biomass (pellet); SPI, Ulva, and Caulerpa refer to *Spirulina platensis*, *Ulva reticulata*, and *Caulerpa lentillifera*, respectively. The suffix -LBA indicates fermentation with *Lactobacillus acidophilus* LB, while MRS-LBA refers to the bacterial control cultured in MRS medium.

The *L. acidophilus* LB strain was initially cultured in 250 mL Erlenmeyer flasks containing 100 mL of MRS medium under static conditions at 35 °C. After 48 h of incubation, the bacterial density reached approximately $1.6 \times 10^7$ CFU/mL (OD$_{600}$ ~0.85). The culture was then transferred to centrifuge tubes and centrifuged at $4,000 \times g$ for 3 min to collect the bacterialpellet. The pellet was washed twice with sterile 0.9% (w/v) NaCl solution to remove residual components of the MRS medium, and finally resuspended in the same solution.

The resulting cell suspension was inoculated into media containing *Spirulina platensis*, *Ulva reticulata*, or *Caulerpa lentillifera* at a ratio of 15% (v/v), and incubated statically at 35 °C for 4 days. Bacterial growth was evaluated by determining the number of viable cells (CFU/mL) using the serial dilution and plate count method on MRS agar. The results showed that the viable cell densities were as follows: $1.6 \times 10^7$ CFU/mL in MRS medium, $4.74 \times 10^8$ CFU/mL in *S. platensis* medium, $4.1 \times 10^8$ CFU/mL in *U. reticulata* medium, and $2.71 \times 10^8$ CFU/mL in *C. lentillifera* medium.

Culture media containing individual types of *S. platensis* biomass, *U. reticulata* seaweed, or *C. lentillifera* extract were prepared as described above, each with a volume of 200 mL. Following the 4-day incubation, the cultures were harvested, and the pellets and supernatants were separated by centrifugation at $1,800 \times g$ for 10 min. The pellets consisted of a mixture of bacterial biomass and residual culture substrates. The pellets were dried at 60 °C, and the supernatants were concentrated using rotary evaporation at 60 °C and 60 rpm (Hei-VAP Core ML/G3 XL; Heidolph, Schwabach, Germany) to obtain crude extracts.

Both the dried biomass and the concentrated supernatants were subsequently analyzed for compound composition using liquid chromatography-high resolution mass spectrometry-quadrupole-time-of-flight (LC-HRMS QTOF) analysis. The positive control in this experiment was *L. acidophilus* LB cultured in MRS medium under identical conditions.

## LCMS QTOF analysis

Approximately five mg of each sample was weighed and dispersed in 1,000 μL of methanol (MeOH). The mixture was vortexed for 2 min, sonicated for 20 min at room temperature, and vortexed again for 1 min. Following centrifugation at $18,000 \times g$ for 5 min, the supernatant was filtered through 0.2 μm filters and transferred to LC vials for analysis. For LC-HRMS analysis, an ACQUITY UPLC I-Class Plus system coupled to a high-resolution Xevo G3 ESI/QToF mass spectrometer (Waters Corporation) was utilized. Chromatographic separation was performed on an ACQUITY UPLC BEH C18 column (130 Å, 1.7 μm, 2.1 mm × 50 mm). The mobile phase consisted of solvent A (H$_2$O + 0.1% v/v formic acid) and solvent B (acetonitrile + 0.1% v/v formic acid). The gradient

elution program began with 20% (v/v) solvent B for 2 min, increased linearly to 95% (v/v) B over 10 min, maintained at 95% (v/v) B for 5 min, returned to 20% (v/v) B over 3 min, and re-equilibrated for 2 min before the next injection. The flow rate of the mobile phase was set at 0.3 mL/min, and the injection volume was five µL. The QToF was operated in MSe continuum mode, scanning from m/z 100 to 1,500 with a scan time of 0.100 s. In low-energy mode, a cone voltage of 6 V was applied, while in high-energy mode, the voltage ramped from 15 to 40 V. Additional settings included a capillary voltage of 3.00 kV (positive mode), source temperature at 120 °C, desolvation temperature at 450 °C, cone gas flow at 30 L/h, and desolvation gas flow at 950 L/h for electrospray ionization (ESI).

The acquisition was performed under the control of MassLynx 4.2 software, and the obtained raw files were processed, visualized, and reported using UNIFI software (Waters Corporation). To dereplicate known compounds in the sample, the QTOF-based chromatogram was processed by UNIFI software using the Waters Traditional Medicine Library (consisting of 6,308 compounds) (Waters Corporation). The detector count of each peak was used to represent the relative intensity of each compound. Comparisons of signal intensities across different metabolites are only qualitative and should be interpreted as relative rather than absolute. Raw peak areas were directly used for statistical analysis without normalization, as all injections were performed under identical conditions and QC samples confirmed minimal technical variation (RSD < 0.3). Metabolite annotation was performed by matching accurate mass and MS/MS fragmentation patterns against publicly available databases. All identifications were assigned level 2 (putatively annotated compounds) according to the Metabolomics Standards Initiative (*Sumner et al., 2007*), as no authentic reference standards were analyzed under the same conditions.

### Statistical analysis

Multivariate statistical analysis methods were employed to investigate untargeted metabolite profiles (*Chatfield & Collins, 1980*; *Cooley & Lohnes, 1971*; *Morrison, 1967*). PCA, Pearson correlation, and the Kruskal–Wallis test were used to analyze data (*Pearson, 1895*; *Kruskal & Wallis, 1952*). PCA reduced dimensionality and visualized sample clustering. Pearson correlation assessed relationships between metabolites, while the Kruskal-Wallis test compared metabolite levels across groups without assuming normality. All analyses were conducted in R (version 4.4.2, the R Foundation) using *prcomp* for PCA, *cor* for correlation, and *kruskal.test* for Kruskal–Wallis (*R Core Team, 2024*).

LC-QTOF analysis was carried out as a single run, whereas all other experimental measurements were performed in technical triplicates.

## RESULTS

### Total compound content and distribution

LC-QTOF analysis identified a total of 151 compounds across all samples. Due to variation in compound distribution, 11 compounds that appeared in at least seven different samples were selected for further statistical analysis.

The total content of variables in all 11 samples was analyzed. Methyl lucidenate Q was the most abundant compound, with a concentration of 5.56 million units, followed by 6′-O-trans-p-coumaroyloleuropein (1.69 million units) and 6-gingerol (761,075 units). Other compounds, such as sulfoorientalol C (441,188 units) and Nigakihemiacetal F (176,297 units), exhibited moderate levels, while auraptenol (140,434 units), cholic acid (125,481 units), and andrographatoside (112,378 units) were found at lower concentrations. The lowest detected compound was tetratriacontanamine, with only 67,535 units.

Coefficient of variation (CV) analysis showed that Methyl lucidenate Q (CV = 0.75) was relatively evenly distributed across samples. Cholic acid (CV = 0.92) and andrographatoside (CV = 0.97) also had relatively even distributions. In contrast, compounds such as 6′-O-trans-p-coumaroyloleuropein (CV = 1.53) and nigakihemiacetal F (CV = 1.42) displayed high concentrations in specific samples, leading to considerable variability.

### Principal component analysis (PCA)

The PCA results (Fig. 1) revealed that PC1 accounted for 44.85% of the variance, while PC2 explained 31.88%, with a cumulative variance of 76.73%. Tetratriacontanamine and sulfoorientalol C contributed significantly to PC1, whereas 6-gingerol and andrographatoside were major contributors to PC2. The loading analysis in PCA identified tetratriacontanamine (0.40), sulfoorientalol C (0.38), and 6′-O-trans-p-coumaroyloleuropein (0.31) as significant contributors to PC1. In PC2, 6-gingerol (0.47), auraptenol (0.47), and daturametelin B (0.46) played crucial roles.

### K-means clustering

K-means clustering analysis with $K = 3$ identified three distinct clusters (Fig. 2). Cluster 1 included the B.Spi, B.Caulerpa, S.Caulerpa, S.SPi-LBA, B.Caulerpa-LBA, and S.MRS-LBA. Cluster 2 contained only the sample B.Ulva-LBA, while Cluster 3 comprised the samples S.Spi, B.Ulva, S.Ulva, B.SPi-LBA, S.Ulva-LBA, S.Caulerpa-LBA, and B.MRS-LBA.

### Kruskal–Wallis analysis

The Kruskal–Wallis test (Fig. 3) identified seven compounds with $p$-values less than 0.05, indicating statistically significant differences among clusters. These include nigakihemiacetal F, songorine, 6-gingerol, tetratriacontanamine, cholic acid, 6′-O-trans-p-coumaroyloleuropein, and sulfoorientalol C. Among these, 6-gingerol exhibited the highest discriminative power.

### Pearson correlation analysis

Pearson correlation analysis (Fig. 4) identified several statistically significant relationships among the variables ($p < 0.05$). A strong positive correlation was observed between auraptenol and daturametelin B ($r = 0.99$), while cholic acid and tetratriacontanamine showed a strong negative correlation ($r = -0.91$). 6-Gingerol exhibited moderate positive correlations with auraptenol ($r = 0.66$) and Daturametelin B ($r = 0.60$). Methyl lucidenate Q showed no strong positive correlations with other compounds but displayed moderate to strong negative correlations with nigakihemiacetal F ($r = -0.73$) and songorine ($r = -0.84$).

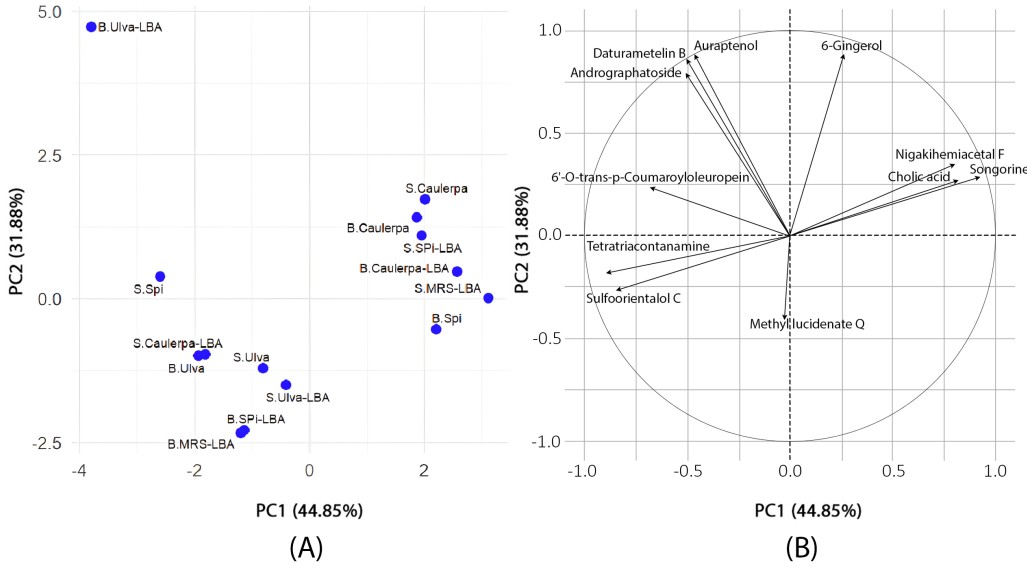

**Figure 1** **Principal component analysis.** (A) The results of PCA to evaluate the differences in metabolite profiles among various samples. Each point on the plot represents an individual sample, labeled to indicate its type and treatment condition. The prefix "S." denotes supernatant samples, while "B." refers to biomass samples. The terms "Ulva," "Spi," and "Caulerpa" correspond to extracts derived from *Ulva reticulata*, *Spirulina platensis*, and *Caulerpa lentillifera*, respectively. The suffix "-LBA" indicates samples cultivated with *Lactobacillus acidophilus* LB, whereas "-MRS" refers to the control samples grown in MRS medium. The two principal components, PC1 (44.85%) and PC2 (31.88%), explain the major portion of variance in the dataset. The distribution and clustering of the points reflect similarities or differences in metabolite composition across sample types and culture conditions. (B) PCA loading plot. The figure displays a PCA loading plot, illustrating the contributions of individual metabolites to the first two principal components (PC1: 44.85%, PC2: 31.88%). The circle in the plot is a *unit circle* with a radius of 1, representing the maximum possible loading value in a correlation-based PCA. Since the PCA was performed using the correlation matrix, the data were standardized so that each variable has a mean of zero and a standard deviation of one. The direction of each arrow indicates how the corresponding metabolite aligns with the principal components, while the arrow length reflects how strongly it contributes to the overall variance. Variables pointing in similar directions are positively correlated; those pointing in opposite directions are negatively correlated.

## DISCUSSION

This study aimed to explore how biochemical differences among three marine-derived substrates, namely *Spirulina platensis*, *Caulerpa lentillifera*, and *Ulva reticulata*, influence the metabolic output of *Lactobacillus acidophilus* LB during fermentation. The untargeted LC-QTOF metabolomics approach revealed clear distinctions in metabolite abundance and distribution across the fermented samples, supporting our hypothesis that substrate composition plays a critical role in modulating microbial metabolism. Multivariate analyses, including PCA and K-means clustering, supported the emergence of distinct chemical profiles, while correlation analysis revealed potential biochemical relationships among key compounds.

Among the detected metabolites, methyl lucidenate Q emerged as the most abundant and consistently distributed compound across most samples, except in B.Ulva-LBA. This

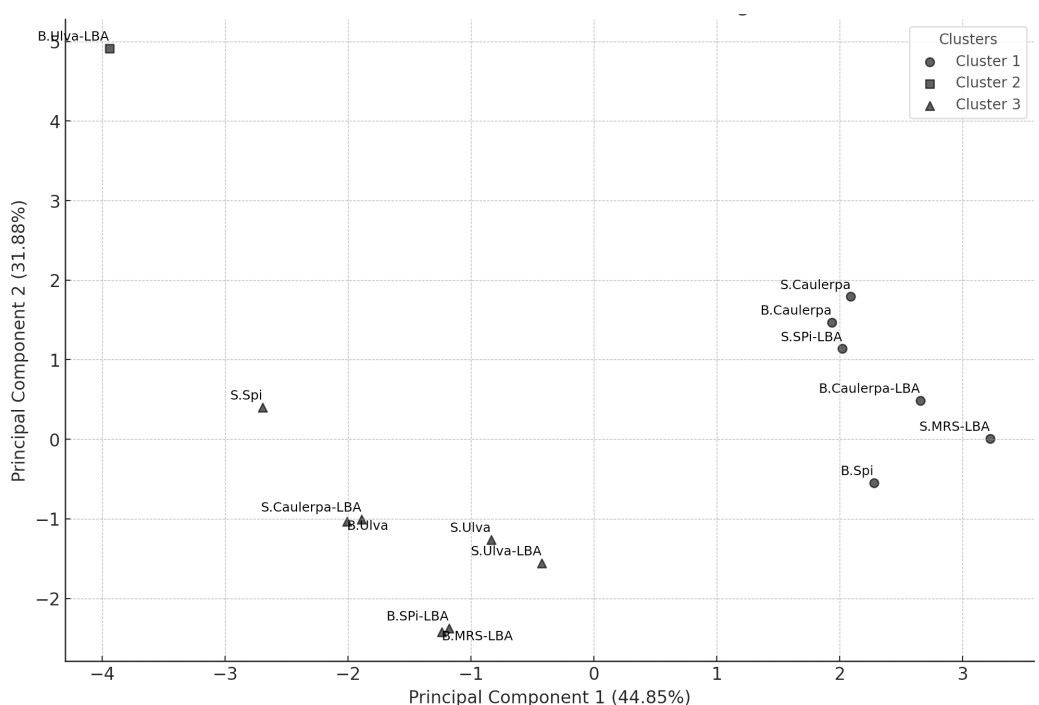

**Figure 2** PCA scatter plot with K-means clustering ($K = 3$) showing the grouping of samples based on their principal components. The horizontal axis represents Principal Component 1 (PC1), explaining 44.85% of the total variance, while the vertical axis represents Principal Component 2 (PC2), explaining 31.88% of the variance. Each point represents a sample, with clusters represented by different symbols: Cluster 1 (l), Cluster 2 (n), and Cluster 3 (p).

triterpenoid, commonly found in *Ganoderma lucidum*, has well-documented antioxidant and anticancer properties (*Chen et al., 2017*; *Iwatsuki et al., 2003*). Although not previously associated with marine algae, its high concentration in *Spirulina* and *Caulerpa*-based fermentations may reflect microbial transformation of structurally related triterpenoid precursors naturally present in these seaweeds. Notably, *Caulerpa lentillifera* is known to contain bioactive terpenes and triterpenes, making enzymatic biotransformation by *L. acidophilus* a plausible explanation. The absence of methyl lucidenate Q in B.Ulva-LBA might indicate either enzymatic degradation or the absence of necessary precursors in *Ulva reticulata*. Although growth kinetics were not measured in this study, the distinct metabolomic patterns and clustering suggest that *L. acidophilus* LB actively proliferated and metabolized in response to different substrates. Future studies should include OD600 and time-course CFU for a more comprehensive growth assessment.

6-Gingerol, a well-known compound from *Zingiber officinale* (ginger), is recognized for its antioxidant and anti-inflammatory properties, which can be enhanced through fermentation with *Lactobacillus plantarum* strains (*Kim et al., 2025*). In the present study, it was detected at moderate levels in the fermented seaweed samples, particularly in *Caulerpa*-based media, raising the possibility that microbial processes during fermentation may have contributed to its presence. Furthermore, its strong association with Cluster 3 in

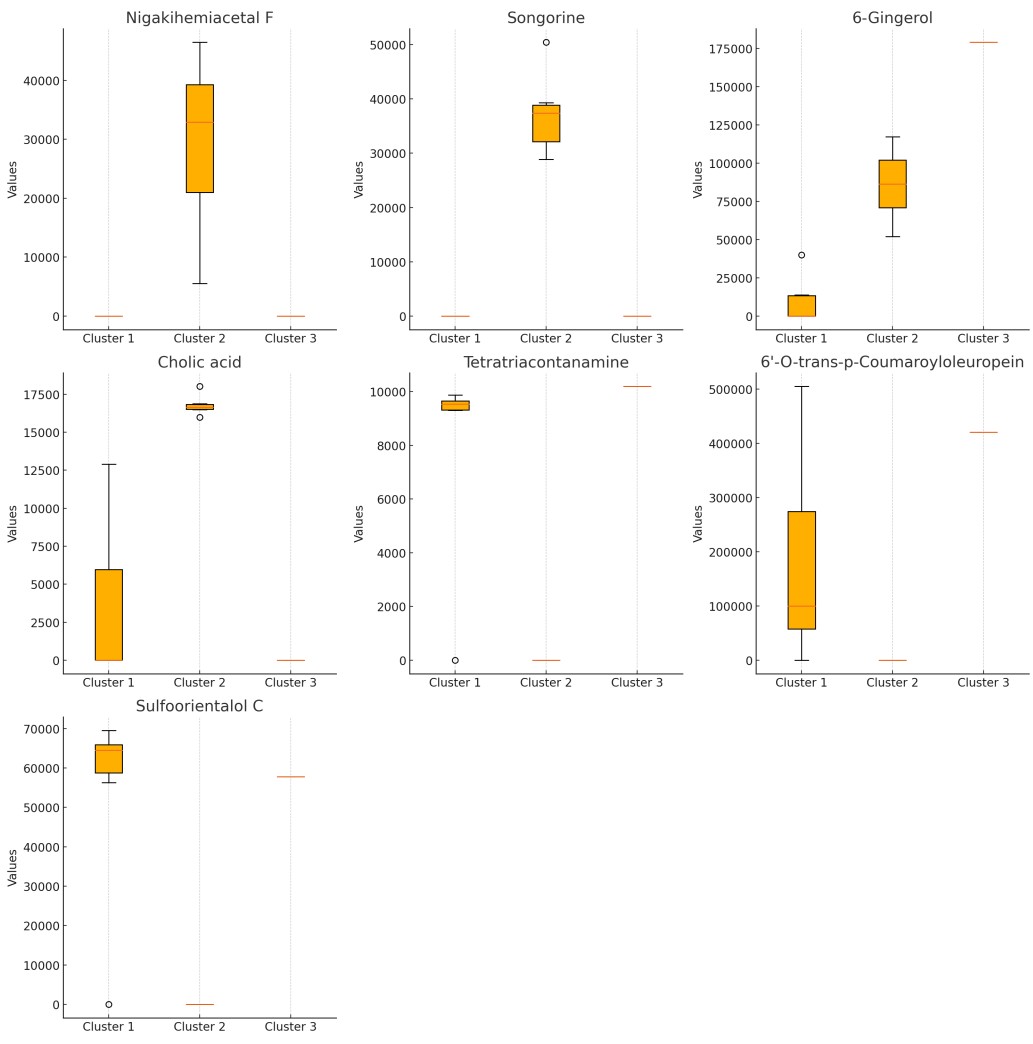

**Figure 3** **Bioactive compounds across clusters obtained from K-means clustering ($K = 3$).** Each subplot represents the compound content (peak area) for a specific bioactive compound across the three clusters. Compounds displayed include nigakihemiacetal F, songorine, 6-gingerol, cholic acid, tetratriacontanamine, 6′-O-trans-p-coumaroyloleuropein, and sulfoorientalol C. The horizontal axis indicates the cluster number, while the vertical axis represents the compound content (peak area).

K-means analysis suggests that specific substrates may favor the biosynthesis or stabilization of this compound under fermentation conditions.

The PCA and K-means clustering provided evidence of distinct metabolic patterns driven by the seaweed substrate. Samples derived from *Spirulina platensis* and *Caulerpa lentillifera* clustered closely, reflecting their similar bioactive composition both rich in proteins, polyphenols, and antioxidants. In contrast, B.Ulva-LBA formed an isolated cluster, suggesting a unique metabolic response. This divergence may stem from *Ulva*'s distinct biochemical profile, including higher levels of flavonoids and polysaccharides (*Gomathi & Anna, 2018*), which may have influenced *L. acidophilus* LB toward alternative

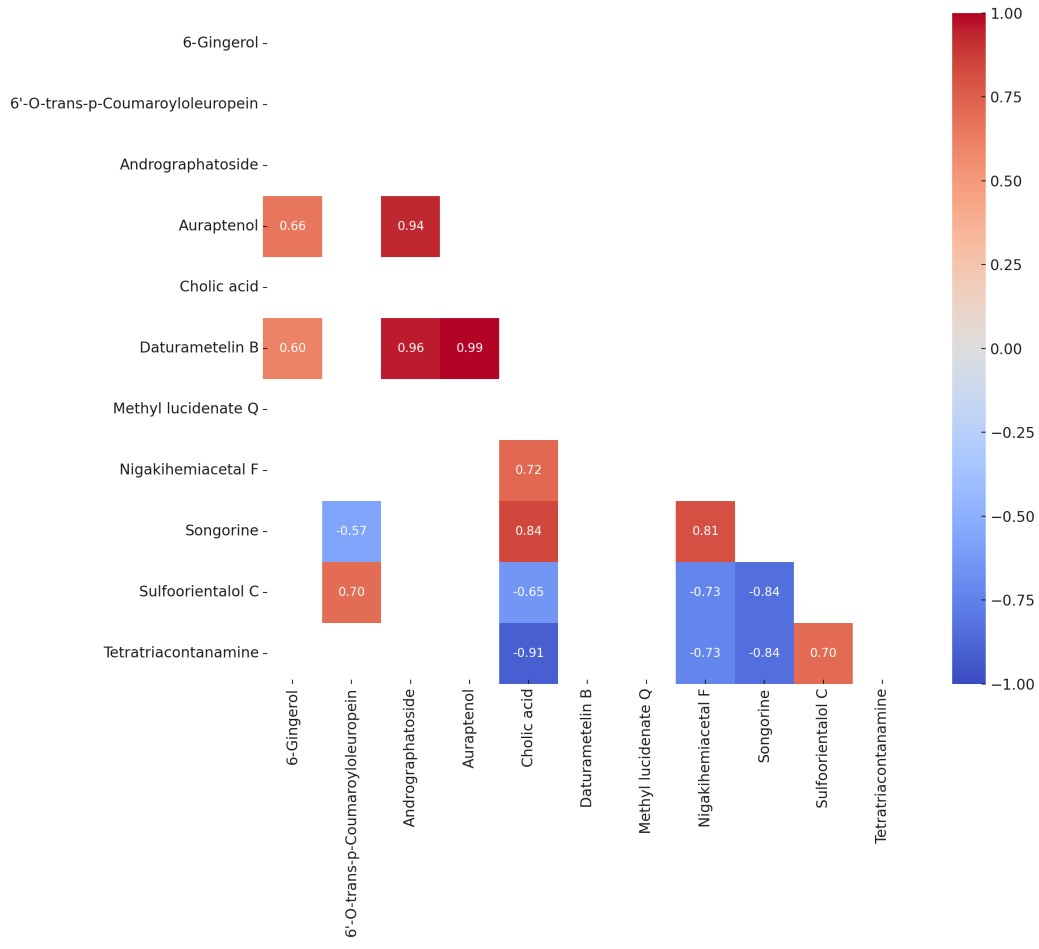

**Figure 4** **Heatmap of Pearson correlation coefficients (*P* value < 0.05) among the bioactive compounds across all samples.** The matrix displays significant pairwise correlations, where positive correlations are shown in shades of red, and negative correlations are shown in shades of blue. The intensity of the color indicates the strength of the correlation.

metabolic pathways. The clustering pattern thus underscores the substrate-dependent modulation of metabolite patterns in the fermented supernatants.

Although S.MRS-LBA was part of Cluster 1, it was relatively isolated compared to the other samples in this cluster. On the PCA plot, both B.Ulva-LBA and S.MRS-LBA were notably distinct from the other samples. The S.MRS-LBA sample, serving as the control medium containing only *Lactobacillus acidophilus* LB, exhibited a distinct chemical profile, explaining its separation from the other samples. B.Ulva-LBA, the solid fraction obtained after centrifuging the co-culture of *Lactobacillus acidophilus* LB with *Ulva reticulata*, displayed unique chemical compositions, even when compared to S.Ulva-LBA, the liquid fraction from the same co-culture. This may suggest that *Ulva reticulata* interacts in a distinctive way with *Lactobacillus acidophilus* LB. The B.Ulva-LBA sample exhibited high concentrations of 6-gingerol and 6′-O-trans-p-coumaroyloleuropein but lacked methyl lucidenate Q. This absence may help explain its distinct position on the PCA plot compared

to other samples. The K-means clustering analysis with $K = 3$ on PCA data revealed clear sample groupings. Cluster 1 consisted of six samples, primarily associated with compounds such as methyl lucidenate Q and sulfoorientalol C. Cluster 2 contained only a single sample (B.Ulva-LBA), characterized by a unique chemical profile. Cluster 3 included seven samples, with significant contributions from compounds like auraptenol and andrographatoside. These results highlight the chemical differences among clusters, providing preliminary insights for identifying the chemical characteristics of the samples.

While PCA and K-means clustering provided a useful overview of sample differentiation and grouping, they did not address the underlying relationships between individual metabolites. Given the biochemical complexity of microbial fermentation systems, it is important to move beyond abundance-based interpretations and investigate how key compounds may interact or co-vary under the influence of different substrates. To address this, Pearson correlation analysis was employed as a complementary approach to uncover potential co-regulation, shared biosynthetic pathways, or mutually exclusive metabolic routes. This allowed for a more nuanced understanding of the microbial metabolic response, particularly in highlighting how specific compounds such as methyl lucidenate Q and 6-gingerol may be functionally linked or inversely related to other bioactive metabolites present in the fermented media.

Pearson correlation analysis among 11 natural compounds revealed several statistically significant relationships, highlighting both structural similarities and potential metabolic linkages. Structurally related pairs, such as daturametelin B–methyl lucidenate Q ($r = 0.99$) and cholic acid–methyl lucidenate Q ($r = 0.91$), shared a common triterpenoid/steroidal polycyclic backbone (*Iwatsuki et al., 2003*; *Akihisa et al., 2007*), which may point to potential coordinated biosynthetic origins. Similarly, pairs originating from the phenylpropanoid pathway, including auraptenol–6′-O-trans-p-coumaroyloleuropein ($r = 0.94$) and auraptenol–6-gingerol ($r = 0.66$), demonstrated functional and structural alignment (*Dixon et al., 2002*; *Vogt, 2010*), with positive correlations indicating possible co-regulation or shared precursors, with potential influence from the polyphenol-rich nature of marine substrates such as *Caulerpa lentillifera*. Notably, auraptenol–daturametelin B ($r = 0.99$) and 6-gingerol–daturametelin B ($r = 0.60$) are consistent with the hypothesis of possible converging biosynthetic pathways under specific fermentation conditions.

In contrast, strong negative correlations such as methyl lucidenate Q-nigakihemiacetal F ($r = -0.73$) and methyl lucidenate Q-songorine ($r = -0.84$) suggest biosynthetic divergence or competitive metabolic relationships, possibly due to substrate competition or mutually exclusive pathways. The lack of strong positive associations involving methyl lucidenate Q further implies its biosynthesis may proceed through an isolated or highly regulated route in *Lactobacillus acidophilus* when utilizing marine-derived matrices. Similarly, the pronounced inverse correlation between tetratriacontanamine-cholic acid ($r = -0.91$) reflects divergent metabolic demands in bacterial processing of lipid-related molecules. The pair nigakihemiacetal F-daturametelin B ($r = 0.72$), though belonging to different classes (quassinoid and withanolide), both exhibit complex polycyclic triterpenoid frameworks commonly associated with bioactive medicinal compounds (*Chen et al., 2015*), suggesting overlapping metabolic routes or complementary ecological functions.

Importantly, daturametelin B and methyl lucidenate Q are pharmacologically valuable compounds with documented immunomodulatory, anti-inflammatory, and anticancer properties, attributable to their multifunctional triterpenoid scaffolds (*Akihisa et al., 2007*). These correlation patterns not only underscore the influence of structural similarity on co-expression but also illuminate the dynamic metabolic flexibility of microbial systems in response to complex marine substrates, providing preliminary insights for the future optimization of fermentation-based production of high-value bioactives in functional food and pharmaceutical applications. As this correlation analysis was exploratory, results were not corrected for multiple testing; thus, statistical associations should be interpreted with caution. These annotations remain tentative and require validation with authentic standards or MS/MS confirmation.This study has some limitations. Each sample was analyzed once by LC-HRMS QToF without internal standards or targeted MS/MS, so metabolite identities remain putative (MSI level 2). Only 11 of 151 detected metabolites (present in $\geq 7$ samples) were statistically examined without multiple-testing correction, potentially increasing the risk of false positives. Analyses such as PCA, k-means clustering, and Pearson correlations describe associations only and do not imply causation. Growth was measured at a single endpoint (CFU/mL after 4 days) rather than by full growth curves, limiting inference on kinetics or substrate utilization. Matrix composition varied between species and fractions, introducing possible matrix effects. Finally, no bioactivity assays were performed; any pharmacological interpretation is literature-based. Future studies should include replicates, isotope-labeled standards, randomized injections, targeted MS/MS validation, multiple-testing control, and functional assays to confirm biological significance.

## CONCLUSION

The observed modulation of metabolite profiles by different seaweed extracts highlights the potential of fermentation-based strategies to valorize marine biomass. By selecting optimal microbial-substrate pairings, key bioactive compounds such as 6-gingerol or methyl lucidenate Q could be selectively enhanced for use in nutraceuticals, functional foods, or cosmeceuticals, pending further bioactivity validation. Nevertheless, this study has limitations: the small sample size reduces generalizability, and metabolite identities were based on tentative annotation without MS/MS confirmation. Future studies should incorporate larger datasets, use authentic standards, and perform functional bioassays to validate the pharmacological relevance of the detected compounds. Moreover, integrating genomic or transcriptomic data would further clarify the microbial pathways responsible for these biosynthetic transformations.

## ACKNOWLEDGEMENTS

We thank Prof. Chu Hoang Ha for providing scientific guidance and valuable advice during the experimental design and manuscript preparation.

### Funding

This work is financially supported by the Vietnam Academy of Science and Technology Grant No. TĐNSH0.06/22-24. The funders had no role in study design, data collection and analysis, decision to publish, or preparation of the manuscript.

### Grant Disclosures

The following grant information was disclosed by the authors:
Vietnam Academy of Science and Technology Grant No. TĐNSH0.06/22-24.

### Competing Interests

The authors declare there are no competing interests.

### Author Contributions

- Ha Phuong Hoang conceived and designed the experiments, performed the experiments, analyzed the data, prepared figures and/or tables, authored or reviewed drafts of the article, and approved the final draft.
- Thi Minh Nguyen conceived and designed the experiments, performed the experiments, analyzed the data, authored or reviewed drafts of the article, and approved the final draft.
- Tuyet Thi Anh Le conceived and designed the experiments, performed the experiments, analyzed the data, authored or reviewed drafts of the article, and approved the final draft.
- Huong Giang Bui analyzed the data, authored or reviewed drafts of the article, and approved the final draft.
- Ngoc Anh Ho analyzed the data, prepared figures and/or tables, authored or reviewed drafts of the article, and approved the final draft.
- Thu Ngo Thi Hoai analyzed the data, prepared figures and/or tables, authored or reviewed drafts of the article, and approved the final draft.
- Nhat Huy Chu conceived and designed the experiments, performed the experiments, analyzed the data, prepared figures and/or tables, authored or reviewed drafts of the article, and approved the final draft.

### Data Availability

The raw measurements are available in the Supplementary File.

### Supplemental Information

Supplemental information for this article can be found online at http://dx.doi.org/10.7717/peerj.20399#supplemental-information.

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
