# Peer review of "Metabolite profiling from the fermentation of marine-derived extracts by Lactobacillus acidophilus LB"

_PeerJ, doi:10.7717/peerj.20399_

## Round 0.1 · original submission · Major Revisions

· Academic Editor

Major Revisions

Dear Dr. Chu, I ask you to correct and supplement the manuscript in accordance with the reviewers' fundamental comments. I hope that your answers to the reviewers will allow them to approve the manuscript for publication.

Reviewer 1 ·

Basic reporting

The article is clearly presented in a professional tone. The abstract is concise, as are the keywords. What stands out is that the introduction cites relatively old literary sources, with little variety in the review—several of the same author groups are cited repeatedly. Between lines 66–95, there are statements that could not have been conceived by the authors themselves, yet there are no references provided for them. These are three paragraphs entirely unsupported by literature. The introduction does not conclude with a clearly defined research objective.
In the Materials and Methods section, not a single method for reference analysis is cited, which is both absurd and unacceptable. The authors must indicate the sources from which the analytical methods were adopted.
Materials – A detailed description of the algae used should be provided, including composition, certificates of origin, etc. This section also contains some abbreviations that are not explained before being used:
Line 106 – PBS
Line 109 – MRS
Lines 119 and 127 – The centrifugation force should be given in g, as it is in line 135.
Line 120 – What does "saline solution" mean here? Is it NaCl or something else? An explanation should be provided.
Lines 140–142 – Clarify what the percentages refer to: w/w, w/v, or something else, just as was done in line 120.
Results – The results are presented correctly, but I have a few remarks regarding Figure 3, which is difficult to read. It would be better if it were split and labeled as parts A and B, with clear explanations of what is being presented in each.
Although Figure 6 effectively conveys the overall statistical significance of the results, I believe that deviations and multiple comparisons should be shown on each individual figure. This would allow the reader to immediately grasp the statistical significance of the data without being referred to a summary figure later on.
Regarding the textual part of the Results section, I have no comments.
Discussion – The discussion is written in a way that fails to provide explanations for any of the findings. It relies entirely on hypotheses. While it's acceptable to work with hypotheses in a scientific article, I believe some of the findings should be explicitly supported and explained with facts.
The conclusion is well-structured.
In-text references and reference list – Line 42 cites "M. et al., 1993, 1997" – the citation style should be reviewed according to the journal’s requirements. In some instances, authors are cited in this abbreviated format, while in others, they are listed as "Surname, Initial(s) et al., Year." This inconsistency also appears with the authors on lines 47, 49, 57, and others. The entire manuscript should be reviewed for consistent citation formatting. These discrepancies are also reflected in the reference list and must be corrected to match the in-text citations and journal guidelines.
Line 289, 294, 307, 333 - references are missing in the text.
Line 317 - described 9 times in the text but not cited as a reference presented in the reference list.
Line 338 - described 12 times in the text but not cited as a reference presented in the reference list.

Experimental design

The article conveys a strong mathematical-statistical focus. However, based on the title and abstract, the reader is led to expect a study with a biological, biotechnological, or technological orientation. The application of such a wide range of statistical and mathematical methods is certainly interesting, but it should not be the central focus. As it stands, it lacks interpretation of the results in a way that is accessible to a broader audience or a more diverse group of specialists.
In my opinion, the discussion section should take a firmer stance, supported by concrete statements, rather than relying predominantly on hypothetical assumptions.
To align with its title, the article should shift more clearly toward a biological direction. Otherwise, the title itself should be revised to highlight the statistical emphasis of the work. However, this brings into question whether the article would still align with the target audience and scope of the journal.

Validity of the findings

All data are presented correctly, though with an excessive emphasis on statistical analysis. There is novelty in the study, but it does not stand out clearly due to the way it is discussed. The current discussion approach resembles more that of a review article rather than an original research paper.

On the positive side, the conclusion is well written.

Additional comments

The article has its merits, but it requires a thorough revision. The focus should be shifted more clearly toward the biological aspect of the research. The discussion and introduction need to be enriched with more recent and diverse sources, especially in areas where references are currently missing—such as the introduction, Materials and Methods, and to some extent, the Discussion.

All technical issues must be corrected, including:

The formatting and clarity of figures

Citation consistency in the text and reference list

The other specific points mentioned earlier (e.g., undefined abbreviations, missing units, explanation of solutions, etc.)

Overall, the paper should be refined and strengthened to meet the standards of a research article and to ensure it accurately reflects its intended biological scope.

Reviewer 2 ·

Basic reporting

The title of the article appears somewhat misleading. As currently written, it suggests that the study involves the cultivation of seaweeds in the presence of Lactobacillus acidophilus, whereas the actual focus is on the fermentation of seaweed extracts. The authors may consider revising the title to more accurately reflect the content and scope of the study.

The introduction section requires significant improvement to better establish the context of the work. First, the background provided is insufficient. The authors should include more information regarding the known chemical composition of the seaweed species studied, rather than emphasizing biological effects, since this manuscript is centered on analytical chemistry rather than biological assays.

Second, the transition into the third and fourth paragraphs (beginning at line 54 and 60, respectively) is abrupt and confusing. While the authors present useful information on L. acidophilus, the link between this probiotic bacterium and seaweed composition is not clearly established. The current framing may mislead readers into thinking the study involves the cultivation of seaweed influenced by bacterial communities, which is not the case. This study evaluates the effects of LAB on seaweed extracts, not on the living seaweed organisms themselves.

In addition, a substantial portion of the research results (lines 66–94) are presented in the introduction. These three paragraphs contain excessive detail for this section and detract from the clarity of the study’s rationale. The function of the introduction should be to lay the groundwork for the study—not to summarize detailed findings before the methods are described. If the authors wish to include an overview of the study, a concise and focused summary would be more appropriate. As it stands, it is not clear which samples or whose metabolites were analyzed, nor how the data presented in these paragraphs were generated. This confusion is exacerbated by the premature introduction of sample names that have not yet been explained.

Finally, the authors should carefully review the manuscript to ensure that all scientific nomenclature, both biological and chemical, is used correctly and consistently throughout.

Experimental design

The study appears to be an original piece of primary research and aligns with the aims and scope of the journal. The use of Lactobacillus acidophilus in the fermentation of seaweed extracts and the subsequent chemical analysis using LC-MS suggests a potentially valuable contribution to the field.

However, the research question was not clearly defined. The introduction section did not adequately establish the knowledge gap or the specific objectives of the study, making it difficult for readers to understand what the authors aimed to investigate. A stronger foundation could be laid by providing more background on the chemical composition of the seaweed extracts, the potential value of their fermentation (e.g., enhancement of bioactive compounds), or even by introducing the concept of postbiotics. This would help contextualize the rationale for fermenting seaweed extracts and clearly define what hypotheses or questions are being addressed.

Furthermore, the experimental setup was not clearly described. While the LC-MS instrument parameters were reported in reasonable detail, the overall methods section lacks critical information. For example, it is unclear what samples were actually analyzed using LC-MS, how many biological or technical replicates were included, and whether any internal or external standards were used for quantification or normalization. Specifically, the authors mentioned the collection of bacterial pellets. The downstream experiments regarding these pellets were unclear. Furthermore, the manuscript does not describe how the raw LC-MS data were processed—what software or tools were used for peak detection, intensity extraction, or feature alignment prior to statistical analysis. These are important methodological details that are essential for assessing the technical rigor, reproducibility, and validity of the findings.

Additionally, the subsection titled “Evaluation of L. acidophilus LB growth in algal/seaweed-containing media” (lines 114-131) lacks clarity and critical methodological information. It does not actually describe how the growth of L. acidophilus was evaluated. The only information given was that dried bacterial biomass was collected, but no data were provided on growth parameters—such as optical density (OD), CFU/mL, or biomass weight—to substantiate or explain this outcome. The experimental conditions under which the LAB were cultured and transferred are also vaguely described. For instance, while it is mentioned that cultures were transferred to different seaweed-containing media after 48 hours of growth in MRS broth at 35 °C, no information is provided on the culture scale (e.g., flask volume or fermenter size), whether the culture was in exponential phase, or if the experiment was carried out under anaerobic conditions. These omissions make it difficult to assess the biological relevance or consistency of the bacterial treatment across samples.

Finally, relying solely on the supplementary table to explain sample groups and design is insufficient. The main text should include a clear and concise description of the experimental design, including sample grouping and treatment rationale, to facilitate reader understanding without requiring them to consult supplementary materials.

Validity of the findings

Due to the insufficient methodological detail provided, it is difficult to fully assess the robustness and validity of the findings. Most critically, the manuscript does not specify the number or type of replicates (biological or technical), and appropriate experimental controls are lacking or inadequately justified. For instance, the rationale for using MRS medium as a control is unclear. If the goal was to benchmark seaweed-supplemented media against MRS in terms of promoting L. acidophilus growth, this should have been explicitly stated and justified. Alternatively, if the aim was to compare the composition of seaweed extracts before and after fermentation, that experimental focus needs to be more clearly defined. The study currently appears to lack cohesion, particularly in how it shifts between analyzing fermented media and bacterial pellets, without explaining the purpose or implications of doing so.

Another important issue is the absence of information regarding metabolite identification. The manuscript does not mention which databases, spectral libraries, or criteria were used to annotate or identify the metabolites detected by LC-MS. This lack of transparency makes it difficult to assess the reliability of the reported results. For example, there is no summary of the total number of metabolites detected, nor is there an explanation for why only a limited subset of metabolites was highlighted in the figures. Given the stated scan range of m/z 100–1500, one would expect a larger and more complex dataset. The authors should clarify whether the reported metabolites were selected based on intensity, fold change, known bioactivity, or other selection criteria.

Furthermore, the quantitative aspect of the data is unclear. In the subsection titled “Total compound content and distribution” (Lines 165–172), the authors refer to compound content and relative concentration, but there is no explanation of how these values were derived. It is unclear whether any absolute or semi-quantitative analysis was performed, whether internal standards were used, or how peak intensities or areas were normalized or compared across samples and compounds. If quantitation was attempted, the methodology should be described in detail, including units used and normalization procedures.

In the discussion, the authors focused on metabolites commonly found in other plant species but did not comment on compounds typically associated with the seaweed species analyzed in this study. Providing prior knowledge on the typical metabolite profiles of the tested algae/seaweed in the introduction would have greatly improved the contextual relevance of the discussion. Additionally, the last five paragraphs of the discussion section (Lines 224–256) lack references, which weakens the credibility of the interpretations made.

Could the authors also clarify the rationale for performing correlation analysis between metabolites? As currently written, the purpose and utility of this analysis are unclear. Wouldn’t it be more informative to correlate metabolite changes with bacterial growth metrics or biological properties of the extracts before and after fermentation? Clarifying this rationale would greatly improve the interpretability and relevance of this analysis.

That said, the conclusions section appears generally acceptable and in line with the presented results, assuming the aforementioned issues are addressed appropriately.

Additional comments

Figure 3 (PCA analysis) requires improvement in both presentation and clarity. The score plot (left panel) and loading plot (right panel) are presented in unequal sizes, making direct comparison and interpretation difficult. The loading plot, in particular, is hard to read due to its small scale and lack of clear visual emphasis. I suggest that the authors resize the panels for better consistency and readability. Additionally, a clear figure legend is needed to indicate which panel represents the score plot and which shows the loading plot. This will help readers better understand the analytical outcome of the PCA. If appropriate and not overly complex, the authors may consider presenting a biplot instead, as it can offer a more integrated view of both sample distribution and variable contribution.

---

## Round 0.2 · Major Revisions

· Academic Editor

Major Revisions

Dear Dr. Chu, I ask you to take into account all the comments of the respected reviewers. I draw your attention to the fact that figures without statistical data processing (for example, figures 1 and 2) are unacceptable. It is better to give these data in the form of tables or just mention these data in the text of the article.

Reviewer 2 ·

Basic reporting

I appreciate that the authors have revised the manuscript in response to previous comments, including adjustments to the tone, the title, and portions of the introduction and literature review. The revised title is notably more engaging and likely to attract interest from readers in the field. However, the introduction still requires improvement.
The opening sentence - “The demand for environmentally friendly microbial fermentation processes that do not compete with human food sources (e.g., glucose) is steadily increasing” - feels somewhat vague and awkward. It is unclear what specific context or evidence supports this claim, and the phrasing could be more precise. Additionally, while the authors state that non-food biomass sources such as marine algae and seaweed are being explored as alternative carbon substrates, they are in fact working with edible seaweed. This creates confusion and weakens the rationale presented in the first paragraph.
Paragraphs 2 to 4 are clearer and demonstrate some improvement, particularly in paragraph 4, which presents a better linkage to the research rationale. However, paragraph 5 would still benefit from refinement in language and structure to enhance clarity and coherence.
I acknowledge that the objectives of the study are now more explicitly stated. However, I would recommend avoiding the use of rhetorical questions such as: "Do these extracts produce distinct metabolic profiles?" or "Can specific biomarkers or sample clusters be identified using appropriate statistical tools?" These weaken the academic tone and are more suitable for discussion rather than objective statements.
Lastly, the paragraph explaining that PCA and Pearson’s correlation are useful tools for metabolomic analysis seems redundant for the intended audience of PeerJ, who are likely already familiar with these commonly used statistical methods. This section could be shortened or omitted to improve the flow of the introduction.

Experimental design

The authors have made commendable progress in improving the Materials and Methods section, particularly in clarifying the LC-MS analysis workflow. The inclusion of more details regarding instrumentation, chromatographic conditions, and analytical steps contributes greatly to the transparency and reproducibility of the study. I encourage the authors to maintain this level of detail in future revisions.

However, some issues related to the alignment between the stated objectives and the actual experimental design remain. In the revised introduction, the authors frame their study around the hypothesis that biochemical differences among seaweed species may influence Lactobacillus acidophilus metabolism, and they propose to assess bacterial growth and secondary metabolic responses. While this is an interesting hypothesis, it does not appear to be well-supported by the experimental setup.

If the authors intend to study bacterial growth and metabolism in response to various seaweed substrates, the current methodology—particularly how the bacterial pellets were handled—would need to be significantly redesigned. The authors collected bacterial cells by centrifugation, separated them from the culture medium, dried the pellets at 60 °C, and then extracted metabolites. This procedure is not suitable for capturing bacterial metabolic activity. It is well-established in the literature that to study intracellular metabolism accurately, metabolic quenching and preservation protocols must be followed immediately upon harvesting cells, in order to prevent degradation or alteration of labile metabolites. The current method likely altered or destroyed many metabolites of interest, and the drying step at elevated temperature constitutes a treatment in itself.

Therefore, I suggest the authors refrain from making claims regarding bacterial secondary metabolism unless appropriate protocols and controls are implemented. Instead, the manuscript would benefit from a sharper focus on the supernatant (i.e., fermented seaweed extracts), which is where the most informative and robust data appear to reside. The metabolite profiling of the supernatants before and after fermentation represents the most scientifically sound aspect of the study and should be emphasized as the central contribution.

Validity of the findings

First, I would like to commend the authors for their honesty and transparency in their response to reviewers. They clearly acknowledged that the LC-MS analysis was conducted as a single run without technical or biological replicates, stating that the sample types were considered sufficiently distinct (e.g., extract residues vs. liquid extracts). While this justification is appreciated, the absence of replication raises concerns about the reliability and reproducibility of the findings. In untargeted metabolomics studies, even limited replication is generally recommended to account for technical variability and to improve confidence in peak detection and compound comparison.

The authors also clarified that no standard compounds were used for metabolite confirmation. While I understand the limitations, this further underscores the importance of detailed and transparent annotation procedures. If the reported metabolites were consistently detected across distinct sample types and the identifications are supported by high-resolution mass accuracy, this lends some credibility to the findings. However, I would caution that, without reference standards or MS/MS fragmentation confirmation, some annotations—particularly those pointing to compounds typically found in terrestrial plants—should be presented as putative identifications (e.g., Level 2 or 3 according to the Metabolomics Standards Initiative).

Interestingly, the authors’ response introduces a new ambiguity regarding the sample composition. They mention "extract residues and liquid extracts" as separate sample types. This raises questions about the nature and purpose of the centrifuged pellets collected from the fermented cultures. Initially, I understood the pellets to be bacterial biomass, especially in the context of the authors' earlier claims regarding bacterial secondary metabolism. However, if these pellets also contain seaweed residues, this should be clearly explained in both the methods and the discussion. Clarifying whether the pellet extractions were intended to analyze bacterial cells, seaweed debris, or a mixture of both is crucial for interpreting the metabolite origin and biological relevance of the data.

Regarding metabolite quantification, I understand that absolute quantification may not have been the goal, and that relative abundance comparisons are acceptable for exploratory metabolomics. My earlier concern about standards stemmed from wanting to ensure the robustness of metabolite identification, not necessarily quantitation. The authors state that detector counts were used to represent relative intensity and to compare abundance levels across samples. This approach is valid, but I recommend that the authors clarify this methodology in the Materials and Methods section. It would also be helpful to include a brief justification for using peak intensity as a proxy for abundance and to specify whether normalization or scaling was applied prior to statistical comparisons.

One important point I would like to clarify is regarding the interpretation of relative metabolite levels. The authors stated that the “detector count of each peak was used to represent the relative intensity of each compound,” and they appear to compare peak areas across multiple metabolites. This approach is potentially misleading. Because different metabolites have varying ionization efficiencies in the ESI source, peak intensities cannot be directly compared across different compounds without normalization using internal standards or calibration. Therefore, while comparing the relative abundance of the same metabolite across samples is valid, comparing different metabolites' signal intensities should be avoided or interpreted with caution. I recommend that the authors clarify this in the manuscript to avoid overinterpretation of semi-quantitative data.

Reviewer 3 ·

Basic reporting

The article is devoted to the detection and analysis of metabolites released during the fermentation of three different algae extracts (Spirulina platensis, Ulva reticulata, and Caulerpa lentillifera) with Lactobacillus acidophilus. The results of this study have a certain significance for the biotechnology of obtaining natural products with potential biological, pharmacological and therapeutic effects. The article is written in a clear scientific and professional language.
The abstract concisely and accurately reflects the essence of the article.
The introduction presents the context of the study and the problems revealed, describes the hypothesis on which this study is based. At the end, the goal is clearly formulated. References to the literature are sufficient and adequate.
The material and methods are described in detail, containing all the necessary information for reproducing the study.
The results of the research are presented in the form of graphs and figures showing the content of biologically active substances in different groups of samples, the total content of variables in all samples, the results of Principal Component Analysis (PCA), PCA scatter plot with K-means clustering (K=3), Diagrams of the distribution of biologically active substances by clusters, Heatmap of Pearson correlation coeûcients. The textual part of the results is well described.
The discussions generally meet the stated goals, contain separate hypotheses and conclusions obtained personally by the authors. References to the literature are sufficient and adequate. In the paragraph located on lines 243–250, references to the literature are necessary, since information about the biological properties and pharmacological effects of 6-Gingerol is provided there.
The conclusion is well written, clear and concise.

Experimental design

Despite the mathematical and statistical focus of this article, I believe it will be interesting for the target audience of the journal. This study adds to the knowledge about the spectrum of biologically active substances synthesized as a result of microbial fermentation of various substrates. The experiment was conducted at a high technical level. The methods used in the study are described fully and in detail, with sufficient information for repetition.

Validity of the findings

The experimental data are presented correctly, with significant and diverse statistical processing and visualization, which may be one of the positive aspects of the article.
The conclusion is clearly formulated, related to the original research question, and limited to supporting results.

Additional comments

Thanks to careful analysis by previous reviewers, the article was significantly revised and finalized. As a result, the article was in a form suitable for publication.
There is no reference in the text of the article: 402 Samanbay, A.; Zhao, B.; Aisa, H.A. (2018). "A new denudatine type C20-diterpenoid alkaloid from Aconitum sinchiangense W. T. Wang." Natural Product Research, 32(19), 2319–2324.
There is an error in the reference on line 236 (Baosong et al., 2017; Kenji I. et al., 2003).
The references on lines 397 and 362 need style correction.

---

## Round 0.3 · Minor Revisions

· Academic Editor

Minor Revisions

Dear Dr. Chu, I ask you to carefully correct the manuscript in accordance with the reviewer's comments.

Reviewer 2 ·

Basic reporting

The manuscript has improved significantly in tone, clarity, and structure compared to earlier versions. The English is more professional, and the introduction now provides a more appropriate background on the chemical composition of seaweeds, avoiding premature presentation of results.

That said, several issues remain. Please carefully check the nomenclature to ensure that all genus and species names are italicized consistently. I also noticed continued use of the term “co-cultivation,” which is not appropriate for this study, as the experiments involved culturing Lactobacillus in media supplemented with seaweed extracts, not with living seaweeds. Font consistency should also be reviewed; some superscripts and subscripts appear in different fonts. Additionally, scientific formatting should be corrected—e.g., g in centrifugal force (× g) and p in p-values should be italicized. Importantly, sample codes should be avoided in the abstract (e.g., “B.Ulva-LBA”); instead, these should be written out in descriptive terms such as “cell pellets of L. acidophilus cultured in U. reticulata extract-containing medium.”

Finally, while the discussion has improved, there are still passages where speculative interpretations of metabolite origins go beyond the strength of the data and should be toned down.

Experimental design

The most critical weakness lies in the analytical chemistry. Each sample was analyzed only once by LC-QTOF, and the authors acknowledge that the experiments cannot be repeated. This severely limits the robustness and reproducibility of the dataset. While this limitation may not be correctable at this stage, it should be clearly emphasized and carefully considered when interpreting and discussing the results.

Bacterial growth was assessed only by reporting endpoint CFU/mL values. Without growth curves, biomass measurements, or additional timepoints, it is difficult to fully evaluate how well the seaweed extracts supported L. acidophilus growth.

In terms of statistical analysis, only 11 metabolites (out of 151 detected) were subjected to testing, based on the arbitrary criterion of being present in ≥7 samples. Moreover, no correction for multiple testing was applied. These limitations reduce confidence in the statistical conclusions and should be acknowledged more explicitly.

Validity of the findings

Some general trends are supported (substrate-dependent clustering of metabolite profiles, differences between supernatants and pellets). However, the validity of the specific compound identifications and their biological interpretation is highly questionable.

1. Unexpected metabolites: Several highlighted compounds (6-Gingerol, Daturametelin B, Methyl lucidenate Q) are known from terrestrial plants or fungi and not from seaweed or lactic acid bacteria. Without precursor confirmation, spiking experiments, or MS/MS validation, these annotations should be regarded as tentative at best. A more likely explanation is contamination during sample preparation or instrument carry-over.

2. Speculative attribution: The manuscript repeatedly interprets these compounds as microbial transformation products, but no evidence is provided to support this. The issue is made worse in the discussion (lines 308–309), where the authors explicitly name plant and mushroom sources in parentheses (e.g., Datura metel, Ganoderma lucidum). This is misleading and inappropriate, as there is no evidence linking these biosynthetic pathways to Lactobacillus or to the substrates used.

3. Correlation ≠ causation: The Pearson correlation analysis is treated as if it reveals mechanistic insights into biosynthetic pathways. In reality, these are statistical associations and should not be overstated as functional relationships.

4. Conclusions vs. data: While the study reasonably concludes that substrate composition influences metabolite profiles, the leap to therapeutic applications or pharmacological value is unsupported without bioactivity assays or functional validation.

Additional comments

This revision represents clear progress. The tone is improved, the structure is clearer, and the authors have made a genuine effort to respond to reviewer concerns. The statistical analyses are reasonably sound. However, the analytical chemistry issues remain unresolved and are serious. Each sample was analyzed only once by LC-MS, with no internal standards or normalization. This undermines reproducibility and contributes to questionable metabolite identifications, with a real possibility of contamination or database-driven misannotation. These limitations, in turn, lead to overstated interpretations of correlations and metabolite origins.

The authors are encouraged to provide supporting references that demonstrate whether the highlighted metabolites have ever been reported in seaweed extracts or as Lactobacillus-associated products. In addition, it would strengthen the work to highlight metabolites that are more plausibly associated with the substrates studied (e.g., Caulerpin from Caulerpa spp. or other well-established seaweed metabolites). This would make the analytical findings more credible.

For the manuscript to be suitable for publication, the authors would need either additional experimental validation (replicates, MS/MS confirmation, internal standards, growth curves) or a substantial reframing of the discussion, focusing on descriptive metabolomic differences and avoiding speculative claims about metabolite origin or pharmacological significance.

Finally, the pharmacological claims should be toned down, particularly the statement in the abstract that “6-Gingerol showed significant pharmacological relevance.” Unless supported by experimental validation (e.g., bioactivity assays), such claims are not justified. If the authors could include even preliminary biological activity evaluations, this would greatly improve the manuscript. However, if the study is intended to be primarily statistics-oriented rather than biotechnological, the discussion should remain within that scope.

---

## Round 0.4 · accepted · Accept

· Academic Editor

Accept

Dear Dr. Chu, I congratulate you on the acceptance of this article for publication.

Reviewer 2 ·

Basic reporting

The authors have revised the manuscript thoroughly and addressed most of the previous comments to the best of their ability, given their current technological and experimental limitations. The manuscript is now clearly written in professional and fluent English, with improved structure, appropriate terminology, and consistent formatting. Importantly, the authors have explicitly acknowledged their methodological constraints, including the single-run LC-QTOF analysis, lack of internal standards, and limited replicates, in the Discussion section. They now present these as clear limitations rather than overextending the interpretation of their data. This responsible framing enhances the transparency and credibility of the work; I think this revised version now meets the journal’s standards for coherence and clarity.

Experimental design

I have no additional comments on the experimental design. The authors did not perform new experiments, re-analyses, or methodological changes in this revision. The experimental setup therefore remains as previously reviewed.

Validity of the findings

Although most of the experimental content remains unchanged (as expected, since the authors did not repeat or reanalyze their work), the authors have expanded the discussion to acknowledge why certain metabolites typically found in terrestrial plants appeared as major features in their fermented seaweed samples. This revised version somewhat shows an attempt to address the concerns in previous comments/suggestions. However, their explanation, especially regarding possible microbial transformation of structurally related triterpenoid precursors, remains speculative rather than scientifically referenced. I still find the argument rather weak; the authors did not provide concrete examples or mechanistic parallels of known bacterial transformations of algal metabolites that lead to similar structures. I, nevertheless, acknowledge several revision attempts and think the revised discussion now frames the authors' interpretations as tentative. This improvement makes the presentation of the findings more acceptable, even if the biochemical reasoning remains limited in depth.